# Structure and Bioactivity of Intracellular and Extracellular Polysaccharides of *Trametes lactinea* Mycelium

**DOI:** 10.3390/microorganisms12071431

**Published:** 2024-07-14

**Authors:** Bowen Dong, Lu Shen, Mei Yang, Kaitai Yang, Fei Cheng

**Affiliations:** 1Guangxi Colleges and Universities Key Laboratory for Cultivation and Utilization of Subtropical Forest Plantation, College of Forestry, Guangxi University, Nanning 530004, China; 2209392007@st.gxu.edu.cn (B.D.); 18234486451@163.com (L.S.); fjyangmei@126.com (M.Y.); 2Guangxi Key Laboratory of Forest Ecology and Conservation, College of Forestry, Guangxi University, Nanning 530004, China; 3Guangxi Forestry Science Research Institute, Nanning 530002, China; kaitai_y2023@163.com

**Keywords:** *Trametes lactinea*, mycelium fermentation, polysaccharide, structural component, bioactivity

## Abstract

*Trametes lactinea* polysaccharides have a high medicinal value; however, we still know little about the structure and bioactivity of intracellular and extracellular polysaccharides in the mycelial liquid fermentation of *T. lactinea*. This study analyzed the structures of intracellular (IP-1, IP-2, and IP-3) and extracellular (EP-1 and EP-2) polysaccharide components isolated from *T. lactinea* liquid fermentation, as well as investigated their antioxidant, antibacterial, and immunomodulatory properties. The results showed that IP-3 was the only component with a triple-helix structure, while the other four components did not possess this structure. IP3 has a higher molecular weight, flavonoid, and total phenolic content compared to other components. Both intracellular and extracellular polysaccharide components exhibited strong scavenging abilities against ABTS and DPPH radicals. The components showed limited antibacterial effects against four types of bacteria (*Staphylococcus aureus*, *Bacillus subtilis*, *Erwinia carotovora*, and *Escherichia coli*), and were found to be non-toxic to RAW264.7 cells, even promoting cell proliferation. Furthermore, within a specific concentration range, all components enhanced the phagocytic activity of RAW264.7 cells, increased the secretion of NO, TNF-*α*, and IL-6, and demonstrated concentration-dependent effects, with IP-3 displaying the most potent immunomodulatory activity. This study shows a high potential for the development and utilization of polysaccharides derived from the liquid fermentation of *T. lactinea* mycelium.

## 1. Introduction

*Trametes lactinea* is a macro-fungus in the Polyporaceae family and *Trametes* genus, widely distributed around the world, mainly in China, Malaysia, Indonesia, and Australia, living in broadleaf timbers and causing white rot of woods [1]. *T. lactinea* can promote asparagus seed germination and shorten the asparagus growth cycle [2], and its laccase is also used for dye decolorization and degradation [3,4]. Several studies have confirmed that *T. lactinea* secondary products have potential medicinal and healthcare functions. For example, *T. lactinea* extract not only showed significant inhibitory effects on hyaluronidase, lipoxygenase, and xanthine oxidase [5], but also has a hepatoprotective effect [6]. *T. lactinea* extracts also contain a variety of active substances that lower blood glucose; for example, the extract site of ethyl acetate was able to lower blood glucose in type II diabetic mice, and the mechanism of lowering glucose was related to increasing insulin sensitivity, decreasing inflammatory factor levels, and improving hepatic glycogen metabolism [7]. Trametenloic acid B is considered to be one of the main active components of *Corynebacterium macrocephala* [8], with an inhibitory effect on H^+^/K^+^-ATPase activity, an anti-gastric ulcer effect, and also the ability to cause selective inhibition of gastric cancer cell survival and a reduction in cell apoptosis [9]. *T. lactinea* culture supernatants are rich in phenolics and flavonoids that scavenge DPPH, NO, H_2_O_2_, and hydroxyl radical groups, suggesting *T. lactinea* is a natural source of antioxidants [5,10]. Nianwu He [11] divided the polysaccharide substance of *T. lactinea* (TLP) into two fractions, TLP-1 and TLP-2. TLP-2 had a better growth-inhibitory effect on L-02 hepatocytes, but TLP-1 possessed a higher reducing capacity and scavenging activity of DPPH radicals, superoxide radicals, and hydroxyl radicals. And TLP-1 had a stronger growth-inhibitory effect on human hepatoblastoma cells HepG-2, and it had a promotional effect on the apoptosis of HepG-2 cells. Nianwu He ‘s research results suggest that *T. lactinea* polysaccharides, especially TLP-1, may have potential anticancer effects. *T. lactinea* polysaccharides from the Tibetan Plateau have an enhancing effect on immunity in immunocompromised mice [12], as well as a strong scavenging ability of oxygen anion radicals and hydroxyl radical activity [13].

Polysaccharides are complex molecules formed by linking together many monosaccharides. They are known for their abundance of polar hydrophilic groups, making them easily soluble in water but poorly soluble in organic solvents. Fungal polysaccharides are natural compounds found in fruiting bodies, mycelia, and fermentation broth [14]. However, since it is difficult to produce mushrooms from *T. lactinea*, there are few wild resources to satisfy their exploitation. Polysaccharides extracted from mycelial liquid fermentation offer advantages over those from fruiting bodies, such as a shorter growth cycle, simpler extraction process, and suitability for large-scale production [15]; mycelial liquid fermentation is one of the most important tools for the further development and utilisation of *T. lactinea*. In our recent study, two extracellular polysaccharide components (EP-1 and EP-2) and three intracellular polysaccharides (IP-1, IP-2, and IP-3) were isolated from the liquid fermentation of *T. lactinea* mycelium; these polysaccharides exhibited distinct monosaccharide structures. For example, IP-1, IP-2, and IP-3 were all highest in glucose. Both IP-2 and IP-3 contained guluronic acid and glucuronic acid. Mannose was the most abundant extracellular polysaccharide in both EP-1 and EP-2 [16].

The biological activity of polysaccharides is heavily influenced by their structural characteristics, including factors such as molecular weight [17], glycosidic bond type [18], monosaccharide composition [18], and triple-helix conformation [19], which indirectly or directly affect the antioxidant activity of polysaccharides. Polysaccharide structures are known for their complexity, featuring a variety of main chain and side chain connections that easily form complexes with other substances. This complexity presents a challenge in fully understanding the biological functions of polysaccharides. Due to the intricate nature of polysaccharide structures, a combination of multiple structural analyses is necessary in the research process.

Therefore, in order to clarify the structure of intracellular and extracellular polysaccharide components of the liquid fermented mycelia of *T. lactinea* and thus reveal their biological activities, in this study, we analyzed the ultraviolet spectra, infrared spectra, triple-helix structure, relative molecular weight, and polysaccharide morphology of these intracellular and extracellular polysaccharide components to further clarify their structure, and evaluated their antioxidant, antibacterial, and immune-regulating functions. This will provide valuable scientific insights for the development and utilization of polysaccharides derived from the liquid fermentation of *T. lactinea* mycelium.

## 2. Materials and Methods

### 2.1. Fruit Body Collection

Fungal fruiting bodies of *T. lactinea* were collected from the Qipo Forest Farm (108°18’ E, 22°28’ N) in Guangxi Province, China. The forest farm is located in a region with a typical subtropical monsoon climate, characterized by an average annual temperature of 21.6 °C, average annual rainfall of 1304.2 mm, average relative humidity of 79%, and an elevation ranging from 150 to 400 m. The land has a slope of 25 to 30 degrees and is primarily hilly, with typical red soil. In late April 2021, fresh fruiting bodies were collected from *Eucalyptus* stumps in a second-generation *Eucalyptus urophylla* × *E. grandis* plantation within the forest farm. A small portion of the tree stumps, along with the fruiting bodies, were carefully cut using a small handsaw and placed in a sealed plastic bag for transportation to the laboratory.

### 2.2. Fungal Isolation, Purification, and Identification

The fruiting bodies were rinsed with running water. The surface of the fruiting bodies was gently wiped with 75% alcohol. They were then rinsed 2–3 times with sterile water. A portion of the fungal cap was cut into small pieces. A small tissue was inoculated into the center of the culture medium (20 g/L glucose, 5 g/L yeast powder, 1.5 g/L KH_2_PO_4_, 1.0 g/L MgSO_4_, 0.1 μg/L vitamin B_1_, and 20 g/L agar) [16]. The inoculated medium was kept at 25 °C. After the hyphae germinated, a small amount of hyphae from the edge of the colony was taken for further purification to obtain a pure culture. Fungal DNA was extracted from the pure culture, and the fungal ITS sequence was amplified using fungal universal primers ITS4 (5′-TCCTCCGCTTATTGATATATGC-3′) and ITS5 (5′-GGAAGTAAAAGTCGTAACAAGG-3′). The sequencing results were uploaded to the NCBI database for comparative analysis using BLAST. The pure cultures were identified as *T. lactinea* based on the sequencing results.

### 2.3. Liquid Fermentation

A small amount of *T. lactinea* mycelium was added to a 250 mL conical flask containing 100 mL of liquid fermentation medium (20 g/L glucose, 5 g/L yeast powder, 1.5 g/L KH_2_PO_4_, 1.0 g/L MgSO_4_, and 0.1 μg/L vitamin B_1_) [16]. The flask was sealed with a sterile sealing film and placed in an oscillating incubator at 30 °C with a shaking frequency of 150 r/min for 3 days to obtain a seed solution. The seed solution was homogenized using an internal cutting homogenizer to form a suspension. The suspension was added to 100 mL of liquid fermentation medium at a volume ratio of 1:20. The liquid fermentation medium was cultivated in the oscillating incubator at 30 °C for 7 days with a shaking frequency of 150 r/min.

### 2.4. Polysaccharide Extraction

#### 2.4.1. Intracellular Polysaccharide

The mycelium and liquid were separated, the mycelium was rinsed with deionized water 4–5 times, freeze-dried and crushed. A suitable amount of mycelium powder was mixed with deionized water in a ratio of 1:40 and placed at 100 °C for 5 h. This extraction process was repeated four times, and the resulting mixture was centrifuged at 8000 rpm for 5 min to combine the centrifuged extraction liquid. Ethanol without water was added to the extraction liquid in a volume ratio of 1:3 and thoroughly mixed. The mixture was then left to stand at 4 °C for 12 h and subsequently centrifuged at 8000 rpm for 5 min to collect the precipitate [16]. The precipitate was dissolved in deionized water to obtain a crude polysaccharide solution. The solution was further dialyzed using a dialysis bag (8000–12,000 Da) with running water for 24 h, followed by dialysis with deionized water for another 24 h. The solution was concentrated using a RE-52AA rotary evaporator (Yarong Biochemical Instrument Factory, Shanghai, China) and then dried using a LC-10N freeze-drying machine (Li Chen Bang Xi Instrument Technology Co, Shanghai, China) to obtain the dry crude intracellular polysaccharide.

#### 2.4.2. Extracellular Polysaccharide

The mycelial fermentation liquid was first separated using gauze and then subjected to centrifugation at 8000 r/min for 5 min. The resulting liquid was combined and concentrated at a volume ratio of 5:1 using a rotary evaporator at 70 °C [16]. To this concentrated mycelial liquid, three times the volume of 100% ethanol was added and vigorously stirred [20]. The mixture was allowed to stand at 4 °C for 12 h, followed by the same steps as the extraction of intracellular polysaccharides, in order to obtain the dry crude extracellular polysaccharide.

### 2.5. Purification and Component Separation of Polysaccharides

The initially extracted crude polysaccharide was dissolved in deionized water, deproteinized with Sevag reagent (chloroform:n-butanol =1:4) [21], and the resulting mixture was centrifuged at 8000 r/min for 10 min. The polysaccharide solution was dialyzed with dialysis bags (8000~12,000 Da) under running water for 24 h. The polysaccharide was concentrated on a rotary evaporator and lyophilized in a freeze dryer, resulting in the dried intracellular and extracellular polysaccharides. The dried polysaccharides were separated by DEAE-52 cellulose anion-exchange chromatography and dextran gel G-100 chromatography into 5 polysaccharides, 3 intracellular polysaccharides (IP-1, IP-2, and IP-3) and 2 extracellular polysaccharides (EP-1 and EP-2) [16].

### 2.6. Structure Characterization of Polysaccharides

UV spectrophotometry was used to scan the range of 190–400 nm with a scanning interval of 1 nm, using a UV–visible absorption spectrophotometer [22]. Fourier transform infrared spectroscopy was used to scan the range of 4000–400 cm^−1^ with a resolution of 4 cm^−1^, using an infrared spectrophotometer [23]. The detection of triple-helix structures was performed by full-wavelength scanning in the range of 200–700 nm using a UV–visible absorption spectrophotometer [22]. Gel permeation chromatography was used to determine the relative molecular weight [24]. Scanning electron microscopy was used to observe and photograph the surface morphology of the gold-coated conductive tape with a small amount of purified intracellular and extracellular polysaccharide samples [23].

### 2.7. Antioxidant Activity

#### 2.7.1. ABTS Free Radical Scavenging Ability

Test tubes were, respectively, filled with polysaccharide components at concentrations of 0.01, 0.05, 0.10, 0.25, 0.50, 1.00, 2.00, 4.00, and 5.00 mg/mL. Two mL ABTS stock solution (Sigma-Aldrich, St. Lois, MO, USA) was added to each tube and thoroughly mixed. The reaction was conducted in the dark at 25 °C for 20 min, and then measured at 734 nm and recorded as A1. The absorbance of deionized water instead of the samples was recorded as A0, and the absorbance of deionized water instead of the ABTS solution was recorded as A2. Vitamin C (VC) was used as a positive control [25]. The half-maximum inhibitory concentration (IC_50_) was used in the test as a reference standard for antioxidant activity. The scavenging rate was calculated using the following formula:$$Scavenging\;rate=\frac{1-(A1-A2)}{A0}\times 100\%$$

#### 2.7.2. DPPH Free Radical Scavenging Ability

The polysaccharide components (2.0 mL) were respectively added into the test tubes. The concentration gradient was set using the same method as the ABTS measurement. Two mL DPPH (0.2 mmol/L) (Sigma-Aldrich, USA) was added and thoroughly mixed. The reaction was carried out at 25 °C in the dark for 30 min, and then measured at 517 nm and recorded as A1. The absorbance of ethanol instead of the samples was recorded as A0, while the absorbance of ethanol instead of the DPPH was recorded as A2. VC was used as a positive control [26]. The calculation formula was the same as for the ABTS measurement.

#### 2.7.3. Hydroxyl Radical Scavenging Ability

Two mL of polysaccharide components with the same concentration gradient as the ABTS measurement was added in the test tubes, and then 2.0 mL ferrous sulfate (9.0 mmol/L) and salicylic acid–ethanol (9.0 mmol/L) were added, followed by 2.0 mL H_2_O_2_ (10.0 mmol/L). The mixture was allowed to react at 37 °C for 30 min, and then centrifuged at 8000 r/min for 5 min. The supernatant was measured at 510 nm and recorded as A1. A0 was recorded as the absorbance when deionized water was used instead of the samples. A2 was recorded as the absorbance when deionized water was used instead of H_2_O_2_. VC was used as a positive control [27].

#### 2.7.4. Iron Ion Reduction Ability

The polysaccharide components (1.0 mL) with the same concentration gradient as in the ABTS measurement were added to the test tubes, followed by 6.0 mL of FRAP reagent. The mixture was allowed to react at 30 °C for 20 min and centrifugated at 8000 rpm for 5 min. The supernatant was measured at 593 nm and recorded as A1. When deionized water was used instead of the samples, its absorbance was recorded as A0. Similarly, when deionized water was used instead of the FRAP reagent, the absorbance was recorded as A2. VC was utilized as a positive control [28].

### 2.8. Antibacterial Activity

*Staphylococcus aureus*, *Bacillus subtilis*, *Erwinia carotovora*, and *Escherichia coli* were activated and their bacterial solutions were diluted. The antibacterial activity of the polysaccharide components against the four bacteria was evaluated using the diameter of inhibition circle (DIC) of filter paper disks [29]. The filter paper disks are soaked with the antibacterial substance diffused in the agar, and the growth of the surrounding bacteria is inhibited to form a clear blank area (DIC) around the paper sheet, which can be used to judge the degree of sensitivity to the inhibitory substance based on the size of the inhibitory circle [30,31]. Filter paper disks soaked in sterile water were used as the blank control, and kantrex was used as a positive control. An inhibition zone diameter greater than 15 mm was considered highly sensitive, 10–15 mm was considered moderately sensitive, 7–10 mm was considered lowly sensitive, and less than 7 mm was considered insensitive. The minimum inhibitory concentration (MIC) refers to the minimum concentration required to exert an inhibitory effect, and the smaller the value, the stronger the inhibitory effect [32,33]. The minimum inhibitory concentration (MIC) was determined using the resazurin microdilution method [34].

### 2.9. Immune Regulatory Activity

The polysaccharide components were then dissolved in DMEM medium to create stock solutions (5.0 mg/mL). From the stock solutions, the samples were prepared in DMEM medium to create a concentration gradient of 50, 100, 200, 400, and 800 μg/mL. Similarly, lipopolysaccharides (LPSs) were dissolved in DMEM medium to prepare a solution (25 μg/mL). To determine the effect of polysaccharide components on the phagocytic function of RAW264.7 cells, the neutral red staining method was used. The blank control was the culture medium, while the positive control was the LPSs. The absorbance was measured at 540 nm. To evaluate the effect of the polysaccharide components on the secretion of NO, TNF-*α*, and IL-6 by the cells, these substances was determined using a NO reagent kit (COIBO BIO, Shanghai, China) and ELISA reagent kit (COIBO BIO, Shanghai, China) [35]. The impact of the polysaccharide components on the cell proliferation activity was determined using the CCK-8 method [36]. An amount of 100 μL of 5 × 10^5^ cells/mL was pipetted into 96-well plates on an ultra-clean bench and incubated at 37 °C in a 5% CO_2_ incubator for 12 h. The supernatant was aspirated off, and 100 μL of the sample solution and 100 μL of the fresh culture solution were added to the blank control wells, and 100 μL of 25 μg/mL LPS solution was added to the positive control wells in accordance with five concentration gradients, and three replicate wells were set up in each group. The 96-well plates were incubated at 37 °C in a 5% CO_2_ incubator for 20 h. After incubation, 1/10 of the volume of culture medium was added with CCK-8 reagent (COIBO BIO, Shanghai, China) in an ultra-clean bench, and the plates continued to be incubated under the same conditions for 4 h. The plates were incubated for 20 h at 37 °C in an ultra-clean bench. The blank control was the culture medium, while the positive control was the LPS. The absorbance was measured at 450 nm.

### 2.10. Statistical Analysis

Microsoft Excel 2019 was used to process all the data, SPSS 20.0 was conducted for One-way ANOVA (New Providence, NJ, USA), and Origin 8.5 was used for plotting. During the experiment, each group was repeated three times, and the experimental data were expressed as mean ± standard deviation. Statistical differences were set at *p* < 0.05.

## 3. Results

### 3.1. Polysaccharide Structure

#### 3.1.1. UV Spectra

No characteristic absorption peaks were observed at 260–280 nm for the five polysaccharide fractions, suggesting the absence of impurities such as proteins and nucleic acids in the IPs and EPs. Furthermore, the absorption curve exhibited a gradual decline beyond 300 nm, with no detectable absorption peaks, indicating a successful purification process and high purity of the polysaccharide fractions (Figure 1).

#### 3.1.2. Infrared Spectra

The infrared scanning spectra of intracellular and extracellular polysaccharide components are showed in Figure 2. Strong and broad peaks were observed at 3413 cm^−1^ for IPs, and at 3421 cm^−1^ for EPs. Weak absorption peaks were observed at 2925 cm^−1^ for IP-1, 2896 cm^−1^ for IP-2, 2921 cm^−1^ for IP-3, and 2925 cm^−1^ for EP-1 and EP-2. The absorption peaks observed at 1656 cm^−1^ for IP-1, 1650 cm^−1^ for IP-2, 1633 cm^−1^ for IP-3, and 1650 cm^−1^ for EP-1 and EP-2 were due to the asymmetric stretching vibration of C=O bonds. These three absorption peaks were characteristic of polysaccharides. The absorption peaks in the range of 1000–1200 cm^−1^ indicated the presence of pyranose sugars, resulting from the vibration of the C-O-H and C-O-C bonds in the pyranose ring [37]. The absorption peaks observed at 933 cm^−1^ and 854 cm^−1^ for IP-1 indicated the simultaneous presence of *α*- and *β*-glycosidic bonds [38], while the absorption peak at 762 cm^−1^ is generated by the symmetric stretching vibration of the C-O-C backbone of D-glucopyranose [39]. The characteristic peaks observed at 875 cm^−1^ for EPs indicated the presence of *β*-glycosidic bonds. These findings demonstrated that both the IPs and EPs possessed characteristic functional groups of polysaccharides, making them typical polysaccharides.

#### 3.1.3. Triple-Helical Structure

Polysaccharides with a triple-helix structure can form complexes with Congo red solution, and a strong alkaline environment will destroy the triple-helix structure, so that its maximum absorption wavelength will be red-shifted, showing a tendency toward increasing and then decreasing [40]. The absorption wavelength of IP-3 displayed an interesting pattern as the NaOH concentration increased, as shown in Figure 3a. Initially, it increased and then decreased, suggesting that its triple-helix structure underwent conformational changes and ultimately transformed into a single-stranded helix structure. The absorption wavelength of IP-1 and IP-2 consistently decreased with higher NaOH concentrations, aligning with the behavior of Congo red itself. This indicates that IP-1 and IP-2 lacked a triple-helix structure. Figure 3b demonstrates that the absorption wavelength of EPs was indistinguishable from that of Congo red, suggesting the absence of a triple-helix structure in both.

#### 3.1.4. Relative Molecular Weight

The chromatograms of all five components displayed single peaks, indicating that each component had a homogeneous molecular weight (Figure 4). Based on the data presented in Table 1, IP-3 had the largest molecular weight, with a weight-average molecular weight (Mw) of 120,686 Da, a number-average molecular weight (Mn) of 76,839 Da, and a polydispersity index of 1.571. Followed by EP-1, it had a weight-average molecular weight (Mw) of 97,077 Da, a number-average molecular weight (Mn) of 62,800 Da, and a polydispersity index of 1.546. Lastly, IP-1 had the smallest molecular weight, with a weight-average molecular weight (Mw) of 37,210 Da, a number-average molecular weight (Mn) of 25,820 Da, and a polydispersity index of 1.441.

#### 3.1.5. Polysaccharide Morphology

The scanning electron micrographs of intracellular and extracellular polysaccharide components are shown in Figure 5. IP-1 exhibited a mostly smooth surface featuring regular circular holes and a compact sheet-like structure. However, there were some slightly elongated protrusions present. IP-2 displayed a dense sheet-like and irregular geometric structure with a relatively rough surface, along with multiple cracks. IP-3’s surface appeared smooth and showcased numerous protruding small particles and coral-like structures, as well as several branched chain structures. Both EPs exhibited smooth surfaces adorned with honeycomb-like structures. EP-1 specifically displayed large circular hole structures, whereas EP-2 showed rectangular hole structures.

### 3.2. Antioxidant Activity

#### 3.2.1. ABTS Free Radical Scavenging Ability

The scavenging activity of ABTS radicals increased progressively with the polysaccharide concentration from 0.01 to 5 mg/mL, demonstrating a strong dose–effect relationship (Figure 6a,b). Notably, the NaCl elution components (IP-2, IP-3, and EP-2) exhibited superior ABTS scavenging ability compared to the water elution components (IP-1 and EP-1). IP-2 and IP-3 effectively eliminated ABTS radicals, surpassing the performance of IP-1. At 3 mg/mL, both IP-2 and IP-3 achieved an ABTS removal rate close to 100%, while IP-1 showed an 87.98% removal rate at 5 mg/mL. EP-2 had a higher ABTS radical scavenging ability compared to EP-1. At 2 mg/mL, the ABTS removal rate by EP-2 was equivalent to VC, while EP-1 exhibited an 84.92% removal rate at 5 mg/mL. EP-2 possessed the strongest ABTS scavenging ability, with an IC_50_ value 19.69 times higher than VC (Table 2). Following closely was IP-2, with an IC_50_ value 20.03 times higher than VC. Conversely, EP-1 exhibited the weakest ABTS scavenging ability, with an IC_50_ value 70.34 times higher than VC.

#### 3.2.2. DPPH Free Radical Scavenging Ability

The NaCl elution components (IP-2, IP-3, and EP-2) exhibited higher DPPH radical scavenging ability compared to the water elution components (IP-1 and EP-1) (Figure 6c,d). When the polysaccharide concentration exceeded 3 mg/mL, both IP-1 and IP-3 displayed a tendency to stabilize the DPPH radical scavenging effect. At 5 mg/mL, the DPPH scavenging rates for IPs were 20.33~97.35%. EP-2 exhibited superior DPPH radical scavenging ability compared to EP-1. Within 0.01–3 mg/mL, EP-2 demonstrated a linear increase in DPPH scavenging rate. As the concentration exceeded 3 mg/mL, the scavenging effect gradually stabilized. At 5 mg/mL, the DPPH scavenging rates for EP-1 and EP-2 were 30.5% and 98.06% respectively. Notably, EP-2 exhibited the strongest DPPH scavenging ability (Table 2), with an IC_50_ value 23.49 times that of VC, while IP-1 had the weakest scavenging ability, with an IC_50_ value 237.85 times that of VC.

#### 3.2.3. Hydroxyl Radical Scavenging Ability

Compared to VC, all components exhibited some level of scavenging abilities, albeit weaker. Notably, the NaCl elution components (IP-2, IP-3, and EP-2) demonstrated a higher hydroxyl radical scavenging ability compared to the water elution components (IP-1 and EP-1) (Figure 6e,f). IP-3, among the IPs, displayed the strongest hydroxyl radical scavenging ability. At a concentration of 5 mg/mL, the scavenging abilities of the IPs stabilized at 12.48%~25.37%. The EPs had similar scavenging abilities, with EP-2 slightly surpassing EP-1. At 5 mg/mL, the scavenging rates of EP-1 and EP-2 were 10.79% and 15.84%, respectively. IP-3 exhibited the strongest scavenging ability (Table 2), with an IC_50_ value 21.80 times that of VC, while EP-1 had the weakest scavenging ability, with an IC_50_ value 44.64 times that of VC.

#### 3.2.4. Iron Ion Reduction Ability

Compared to VC, all five fractions showed some ability to reduce iron ions, although their reduction ability was relatively weak. Among these fractions, the NaCl elution fractions (IP-2, IP-3, and EP-2) exhibited a higher iron ion reduction ability compared to the water elution fractions (IP-1 and EP-1) (Figure 6g,h). The iron ion reduction ability of the IPs increased with increasing concentration within 0.01–4 mg/mL. At 4 mg/mL, the reduction ability of IPs was measured at 15.63 μmol/L~48.72 μmol/L. Beyond 4 mg/mL, these three fractions reached stabilization. The reduction ability of EP-2 was found to be superior to that of EP-1. At 5 mg/mL, the reduction ability of EP-1 and EP-2 was measured at 14.52 μmol/L and 42.97 μmol/L, respectively. IP-3 exhibited the strongest iron ion reduction ability, with an IC_50_ value 17.82 times higher than that of VC. Conversely, EP-1 displayed the weakest reduction ability (Table 2), with an IC_50_ value 57.26 times higher than that of VC.

### 3.3. Antibacterial Activity

The antibacterial effects of polysaccharide components are shown in Table 3. Among the components, IP-3 exhibited the largest inhibitory zone diameter (8.57 mm) against *B. subtilis*, followed by IP-2 (7.33 mm), both showing low sensitivity. Similarly, IP-3 demonstrated a higher inhibitory effect with an inhibitory zone diameter of 8.06 mm against *S. aureus*, while the other four components had no sensitivity. All five polysaccharide components showed insensitivity towards *E. coli*, with IP-2 (4.32 mm), IP-3 (3.87 mm), EP-2 (3.19 mm), EP-1 (3.03 mm), and IP-1 (3.02 mm) exhibiting decreasing inhibitory effects. The inhibitory effect of the five components on *E. carotovora* was also found to be insensitive, with IP-3 (5.85 mm), IP-2 (5.36 mm), EP-2 (4.42 mm), EP-1 (4.04 mm), and IP-1 (3.94 mm) exhibiting decreasing inhibitory effects. Notably, the minimum inhibitory concentration of IP-2 and IP-3 against the four bacteria was less than 5.000 mg/mL, indicating a good inhibitory effect on all four bacteria.

### 3.4. Immune Regulatory Activity

#### 3.4.1. Cell Proliferation Activity

Compared to the control (CK), all five fractions demonstrated the ability to promote the proliferation of RAW264.7 cells within the concentration range of 50–800 μg/mL, without causing any cytotoxic effects (Figure 7). Notably, IP-1 exhibited significant cell proliferation activity at 100–200 μg/mL, with a maximum increase of 104.67%. Similarly, IP-2 showed significant proliferation activity at 400–800 μg/mL, with a maximum increase of 113.94%. IP-3 displayed the most pronounced proliferation activity, reaching a maximum increase of 136.27% at 800 μg/mL. EP-1 demonstrated a significantly dose-dependent proliferation activity at 100–800 μg/mL, with a maximum increase of 122.98%. EP-2 also significantly promoted cell proliferation, reaching a maximum increase of 117.56% at 800 μg/mL. Notably, IP-3 showed the strongest proliferation activity, with a relative cell survival rate of 82.59% at 800 μg/mL, comparable to the positive control (LPS).

#### 3.4.2. Phagocytic Ability

Compared to the CK, the five components demonstrated a dose-dependent effect on the phagocytic activity of RAW264.7 cells within the concentration range of 50–800 μg/mL (Figure 7). A significant increase in IP-1 and IP-2 phagocytic activity increased with doses from 100 to 800 μg/mL. IP-3 also displayed significant phagocytic activity within the concentration range, and exhibited a dose-dependent pattern. At 800 μg/mL, the IPs showed the most substantial enhancement of cell phagocytic activity, with OD increases of 28%~54%, compared to the CK. Similarly, a significant difference in phagocytic activity was observed for EP-1 compared to the CK within the concentration range, and a dose-dependent pattern. EP-2 also exhibited increased phagocytic activity with increasing doses, showing a significant difference compared to the CK. At 800 μg/mL, EP-1 and EP-2 displayed the strongest promotion of phagocytic activity, with OD increases of 45% and 39%, respectively, compared to the CK. Notably, IP-3 exhibited the most significant enhancement of phagocytic activity, with an OD value of 93.33% at 800 μg/mL.

#### 3.4.3. Ability of Cells to Secrete NO

Within the concentration range of 50–800 μg/mL, the NO production by RAW264.7 cells increased with the increase in polysaccharide concentration, and there was a significant difference compared to the CK (Figure 7). The five components reached their maximum NO values at 800 μg/mL, ranging from 17.06 μmol/L to 24.62 μmol/L. Notably, IP-3 demonstrated the strongest ability to stimulate NO production, with a NO concentration reaching 85.49% that of the positive control at 800 μg/mL.

#### 3.4.4. Ability of Cells to Secrete TNF-α

Compared to the CK, the five fractions significantly induced the release of TNF-*α* in RAW264.7 cells (Figure 7). Within the concentration range of 50–800 μg/mL, TNF-*α* increased with the increase in polysaccharide concentration, showing a dose-dependent relationship. At 800 μg/mL, the maximum secretion of TNF-*α* was observed for the five polysaccharide fractions (230.38~405.81 pg/mL, amounting to 60.63~106.79% that of the positive CK). At 800 μg/mL, the effect of IP-3 on the secretion of TNF-*α* in RAW264.7 cells was higher than that of the positive CK (LPS).

#### 3.4.5. Ability of Cells to Secrete IL-6

At concentrations ranging from 200 to 800 μg/mL, IP-1 demonstrated a dose-dependent effect on IL-6 secretion in RAW264.7 cells, showing a significant difference compared to the CK (Figure 7). Similarly, both IP-2 and IP-3 exhibited significant differences in IL-6 secretion compared to the CK, demonstrating a dose-dependent effect. At the highest concentration of 800 μg/mL, the three IPs showed maximum secretion levels of IL-6 (355.46~747.69 pg/mL). The EPs also displayed significant differences in IL-6 secretion compared to the CK, exhibiting a dose-dependent effect. At 800 μg/mL, EP-1 and EP-2 reached maximum secretion levels of IL-6 at 699.13 pg/mL and 475.91 pg/mL, respectively. Notably, IP-3 exhibited the highest ability to stimulate IL-6 secretion, reaching a secretion level equivalent to 90.08% that of the positive control (LPS) at 800 μg/mL.

## 4. Discussion

In another study on the *T. lactinea* polysaccharide extraction process, our team analyzed the chemical composition of the five polysaccharides (IP-1, IP-2, IP-3, EP-1, and EP-2) of *T. lactinea* and came up with the following results: IP-1 consisted of fucose (Fuc), galactose (Gal), glucose (Glc), and mannose (Man). IP-2 and IP-3 both consisted of Fuc, glucosamine hydrochloride (GlcN), Gal, Glc Man, Gulo-glucuronic acid (GulA), and glucuronic acid (GlcA). IP-1, IP-2, and IP-3 all had the highest content of Glc, IP-2 and IP-3 contain GulA and GlcA, and IP-3 had higher levels of GulA and GlcA. EP-1 consisted of Fuc, aminogalactose hydrochloride (GalN), arabinose (Ara), GlcN, Gal, Glc, and Man. EP-2 consisted of Fuc, GalN, Ara, GlcN, Gal, Glc, Man, and Rha. Both EP-1 and EP-2 had the highest content of Man, and none of them contained a glyoxylate component. After isolation and purification, both intracellular and extracellular polysaccharide fractions contained small amounts of flavonoids and total phenols, with IP-3 containing higher amounts of flavonoids and total phenols. IP-2, IP-3, and EP-2 did not contain proteins. IP-3 had the highest content of glyoxylate, with IP-3 > IP-2 > EP-2 >EP-1 > IP-2 [16]. Flavonoids are conjugated to polysaccharides by different graft copolymerization methods and these compounds are called flavonoid-grafted polysaccharides [41]. Liu’s research shows that the conjugation of flavonoids can significantly improve the antioxidant, antimicrobial, antitumor, hepatoprotective, and enzyme inhibition properties of polysaccharides [42]. The excellent biological activity of IP-3 compared to other polysaccharide fractions may be related to its higher flavonoid content.

The polysaccharide components showed good scavenging ability for ABTS and DPPH radicals, but weak scavenging ability for hydroxyl radicals and ferric ion reduction. Studies have shown that low-molecular-weight polysaccharides have higher antioxidant activity, which is consistent with the results of this study [43]. The molecular weight of EP-2 (54,753 Da) is lower than that of EP-1 (97,077 Da), so EP-2 has higher antioxidant activity than EP-1, which is also consistent with the study by Yang [44]. The scavenging ability of the IPs for ABTS and DPPH radicals is as follows: IP-2 > IP-3 > IP-1. This may be because the content of mannose in the monosaccharide composition is positively correlated with the scavenging of ABTS and DPPH radicals [45]. However, due to the extremely low content of glucuronic acid in IP-1 and the higher content of glucuronic acid in IP-3, the antioxidant capacity of IP-3 was higher than that of IP-1, which is consistent with the study by Hu [46]. The antioxidant activity of the fungal mycelial polysaccharides studied by Hao [39] for ABTS radicals was lower than the scavenging activity of intracellular and extracellular polysaccharides of *Trametes versicolor* in this study, while the scavenging activity for hydroxyl radicals and ferric ion reduction was higher than that of this study. The difference in antioxidant activity of these polysaccharides may be closely related to the strain source, culture conditions, and extraction conditions.

This study investigates the immunomodulatory function of intracellular and extracellular polysaccharides from *T. lactinea* using the RAW264.7 cell model. The *T. lactinea* IPs and EPs significantly promoted RAW264.7 cell proliferation, enhanced phagocytic activity, and stimulated the release of NO, TNF-*α*, and IL-6, which might be closely related to their monosaccharide composition. All five polysaccharide components contained fucose, galactose, glucose, and mannose. Relevant studies have shown that the recognition of galactose, glucose, and mannose in polysaccharide components by cell receptors can stimulate the immunomodulatory response of macrophages [47,48].

Infrared spectroscopy revealed that the IPs and EPs contained stretching vibration peaks of O-H bonds, C-H bonds, and C=O bonds, as well as pyranose ring configurations. Previous studies have shown that the antibacterial activity of *Ganoderma lucidum* mycelial polysaccharides is attributed to the presence of *β*-D-glucans [11]. The carboxyl group (C=O) functional group in polysaccharides cannot only provide more lone pairs of electrons to display antibacterial and cholesterol-lowering activities, but also create an acidic environment to promote polysaccharide hydrolysis and exhibit antioxidant activity [40]. Therefore, some functional groups in polysaccharides are closely related to their biological activities.

Studies have shown that the different triple-helix structures of polysaccharides inevitably lead to differences in biological activities [49], and polysaccharides with a triple-helix structure have stronger biological activity than single-stranded helical structure polysaccharides [50]. In this study, only IP-3 contained a triple-helix structure, while other polysaccharides did not. Research has shown that the molecular weight of polysaccharides is closely related to the triple-helix structure, with larger molecular weight polysaccharides (>90 kDa) more likely to form a triple-helix structure. This study’s results are consistent with this theory, as IP-3 has the largest molecular weight of 120,686 Da, making it more likely to form a triple-helix structure. The average molecular weight of the polysaccharides isolated by Zhou from a split-fold mushroom is 2.5 × 10^7^ Da [51], significantly higher than 90 kDa, and the Congo red test indicates the presence of a triple-helix structure. Studies have shown that the molecular weight, polymerization degree, and modification of polysaccharides affect their antibacterial effects [52]. In this study, the high-molecular-weight IP-3 (120,686 Da) had better antibacterial effects than other polysaccharide components, which is consistent with the finding that high-molecular-weight polysaccharides exhibit higher antibacterial activity than low-molecular-weight polysaccharides [53]. Among them, IP-3 had strong antibacterial activity against *B. subtilis*, *S. aureus*, and *P. aeruginosa*, while IP-2 had strong antibacterial effects against *E. coli*. Tang [54] first discovered that the extracellular polysaccharides produced by the liquid fermentation of *Rhizopus microsporus* can significantly inhibit *Staphylococcus aureus*, *Candida albicans*, and *Saccharomyces*, with inhibition zone diameters greater than 10 mm, while the extracellular polysaccharides from *T. lactinea* in this study had an inhibition zone diameter of less than 2 mm against *S. aureus*. Li [35] studied the intracellular polysaccharide NSPG-1 from *Pleurotus ostreatus* and found that it had a certain inhibitory effect on *E. coli*, *B. subtilis*, and *S. aureus*, with minimum inhibitory concentrations of 7.56 mg/mL, 12.54 mg/mL, and 10.42 mg/mL, respectively. In this study, the polysaccharides had higher antibacterial activity against these three bacteria, and the minimum inhibitory concentrations were all less than 10.000 mg/mL.

Significant differences in microscopic morphology were observed between the five fractions of *T. lactinea*. The intracellular polysaccharides were flaky, while the extracellular polysaccharides were reticular. IP-3 had a smooth surface with numerous protruding particles and coral-like structures, with a molecular weight of 120,686 Da. This may be due to the presence of a triple-helix structure and a larger molecular weight, resulting in protruding particles and coral-like structures accompanying the flaky structure. The EPs had smooth surfaces with many reticular structures. Cai [55] speculated that this may be due to the formation of reticular pores caused by sublimation of water during the freeze-drying of polysaccharides. The five fractions exhibited irregular structures, indicating the absence of crystalline structures.

## 5. Conclusions

Five components of polysaccharides were extracted and purified from *T. lactinea* liquid fermentation, intracellular polysaccharides IP-1, IP-2, and IP-3, and extracellular polysaccharides EP-1 and EP-2. Intracellular polysaccharides are more biologically active than extracellular polysaccharides. The IPs and EPs showed strong scavenging abilities against ABTS and DPPH free radicals. The polysaccharide components did not exhibit any toxic effects on the growth of RAW264.7 cells and actually promoted cell proliferation. Each component enhanced cell phagocytic activity and increased the secretion levels of NO, TNF-*α*, and IL-6, with effects dependent on concentration. Among them, IP-3 has the strongest immunomodulatory activity, antimicrobial activity, and highest ability to clear hydroxyl free radicals and reduce iron ions. IP-3 was the only component with a triple-helix structure, while the other four components did not possess this structure. IP3 has a higher molecular weight, flavonoid, and total phenolic content compared to other components. Therefore, it can be seen that larger molecular weight polysaccharides are more likely to form a triple-helix structure, and polysaccharides with a triple-helix structure have stronger biological activity than single-stranded helical structure polysaccharides. Meanwhile, the content of flavonoids greatly affects the biological activity of polysaccharides. These findings provide valuable insights for the development and utilization of *T. lactinea* biological resources.

## Figures and Tables

**Figure 1 microorganisms-12-01431-f001:**
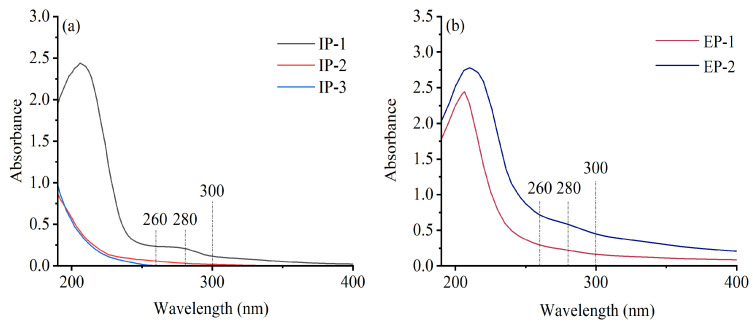
UV scanning patterns of intracellular (**a**) and extracellular (**b**) polysaccharides. IP-1, IP-2, and IP-3 are intracellular polysaccharide components; EP-1 and EP-2 are extracellular polysaccharide components.

**Figure 2 microorganisms-12-01431-f002:**
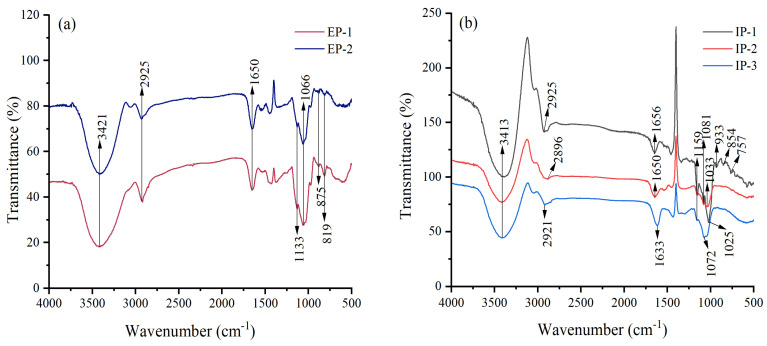
Infrared scanning spectra of extracellular (**a**) and intracellular (**b**) polysaccharides. IP-1, IP-2, and IP-3 are intracellular polysaccharide components: EP-1 and EP-2 are extracellular polysaccharide components.

**Figure 3 microorganisms-12-01431-f003:**
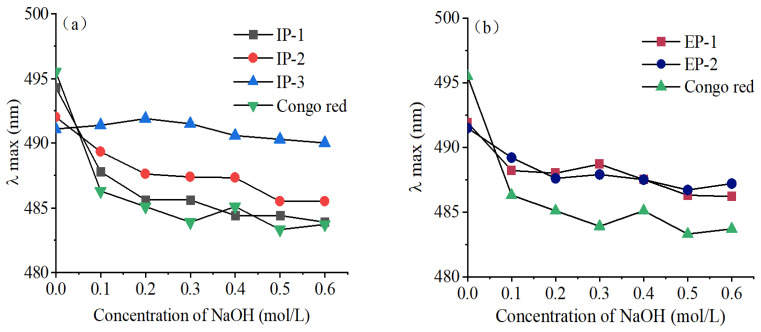
Congo red test for intracellular (**a**) and extracellular (**b**) polysaccharides. IP-1, IP-2, and IP-3 are intracellular polysaccharide components: EP-1 and EP-2 are extracellular polysaccharide components.

**Figure 4 microorganisms-12-01431-f004:**
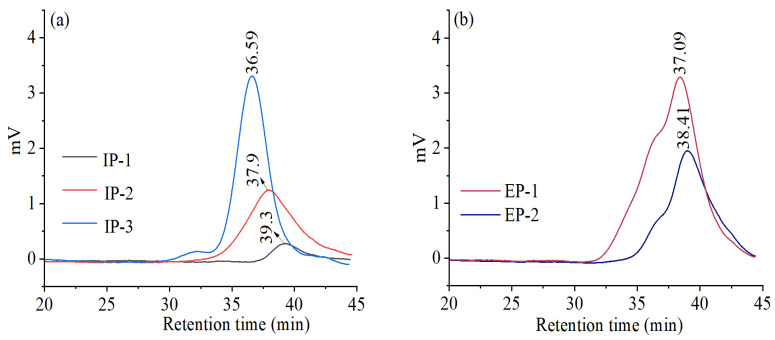
Molecular weight distribution of intracellular (**a**) and extracellular (**b**) polysaccharide. IP-1, IP-2, and IP-3 are intracellular polysaccharide components; EP-1 and EP-2 are extracellular polysaccharide components.

**Figure 5 microorganisms-12-01431-f005:**
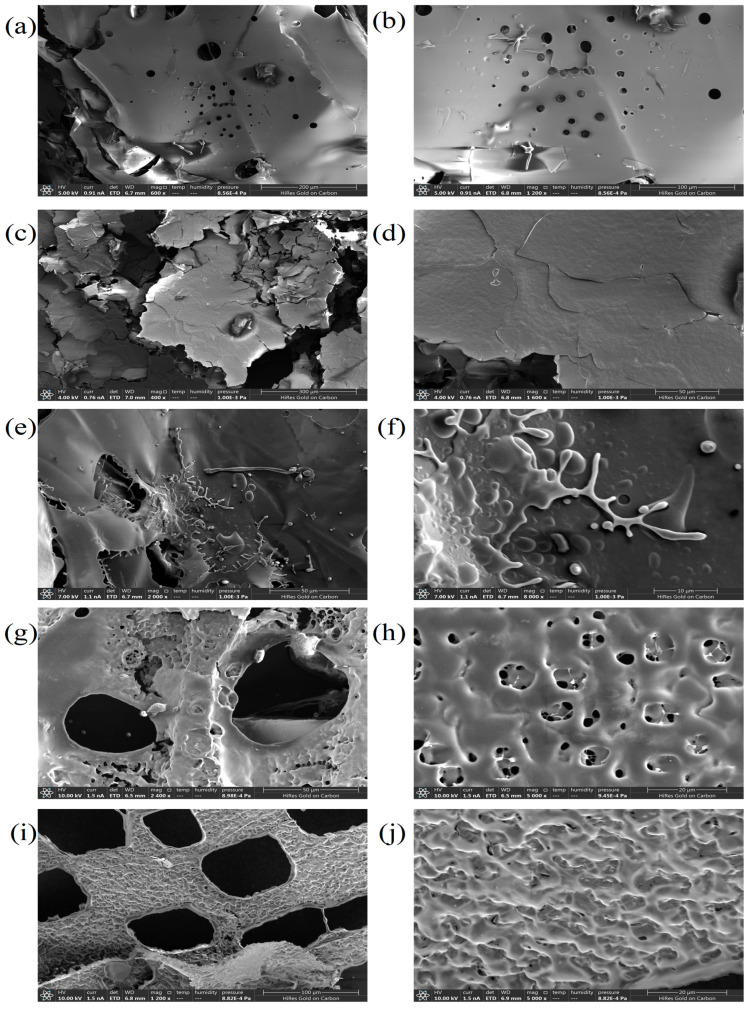
Scanning electron micrographs of IP-1 (**a**,**b**), IP-2 (**c**,**d**), IP-3 (**e**,**f**), EP-1 (**g**,**h**), and EP-2 (**i**,**j**). IP-1, IP-2, and IP-3 are intracellular polysaccharide components; EP-1 and EP-2 are extracellular polysaccharide components.

**Figure 6 microorganisms-12-01431-f006:**
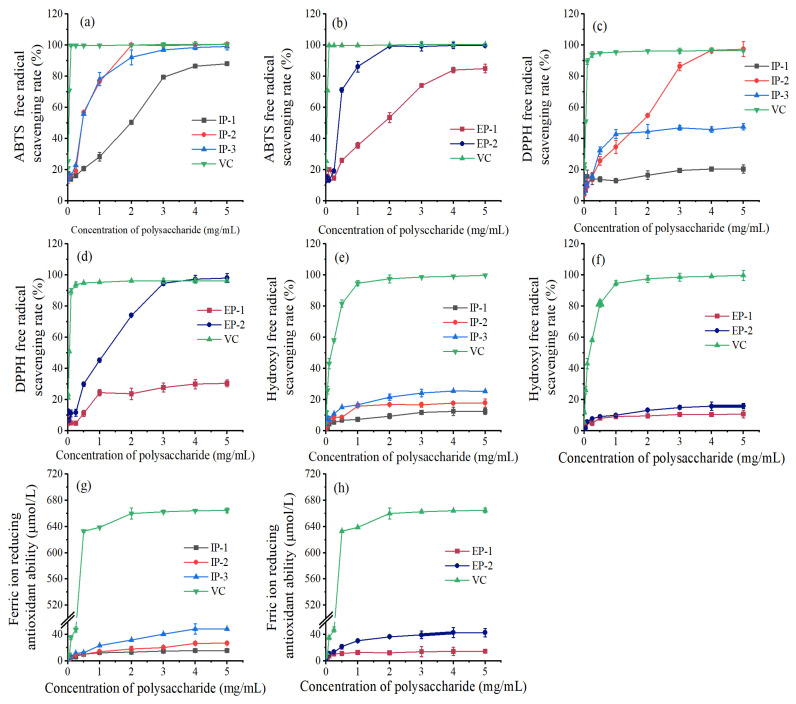
Antioxidant activity of intracellular and extracellular polysaccharides. IP-1, IP-2, and IP-3 are intracellular polysaccharide components; EP-1 and EP-2 are extracellular polysaccharide components; and VC is Vitamin C.

**Figure 7 microorganisms-12-01431-f007:**
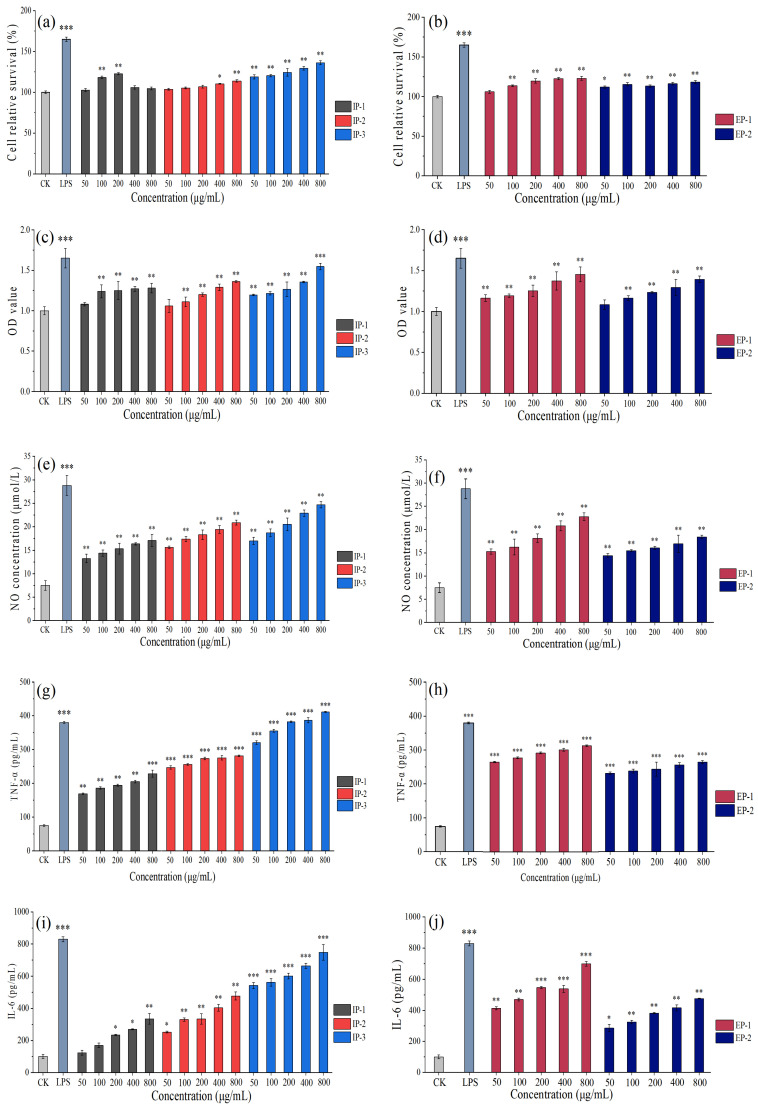
Effect of different concentrations of intracellular (**a**,**c**,**e**,**g**,**i**) and extracellular (**b**,**d**,**f**,**h**,**j**) polysaccharides on the proliferative activity (**a**,**b**), phagocytic ability (**c**,**d**), and ability of cells to secrete NO (**e**,**f**), TNF-α (**g**,**h**), and IL-6 (**i**,**j**) of RAW264.7 cells. IP-1, IP-2, and IP-3 are intracellular polysaccharide components; EP-1 and EP-2 are extracellular polysaccharide components; LPS, lipopolysaccharides; “*” indicates a statistically significant difference (*p* < 0.05), “**” indicates a significant difference (*p* < 0.01), and “***” indicates a highly significant difference (*p* < 0.001) for each concentration gradient compared with the CK.

**Table 1 microorganisms-12-01431-t001:** Relative molecular weights of intracellular and extracellular polysaccharides.

Index	Intracellular			Extracellular	
	IP-1	IP-2	IP-3	EP-1	EP-2
Mw (Da)	37,210	68,276	120,686	97,077	54,753
Mn (Da)	25,820	45,320	76,839	62,800	36,935
Mw/Mn	1.441	1.507	1.571	1.546	1.482

Note: Mw, heavy average molecular weight; Mn, number-average molecular weight; and Mw/Mn, distribution width index.

**Table 2 microorganisms-12-01431-t002:** IC_50_ value of ABTS, DPPH, hydroxyl radical scavenging, and iron ion reduction.

Index	Component	IC_50_ (mg/mL)
ABTS	IP-1	2.016
	IP-2	0.581
	IP-3	0.818
	EP-1	2.040
	EP-2	0.571
	VC	0.029
DPPH	IP-1	14.033
	IP-2	1.710
	IP-3	4.047
	EP-1	6.359
	EP-2	1.386
	VC	0.059
Hydroxyl	IP-1	11.444
	IP-2	9.706
	IP-3	7.977
	EP-1	14.998
	EP-2	10.026
	VC	0.366
Iron ion	IP-1	10.730
	IP-2	7.791
	IP-3	4.223
	EP-1	13.571
	EP-2	4.569
	VC	0.237

**Table 3 microorganisms-12-01431-t003:** Diameter of inhibition circle (DIC) and minimum inhibitory concentration (MIC) of intracellular and extracellular polysaccharides against test strains.

Index	Polysaccharide Component	Bacterial Strain			
		*B. subtilis*	*S. aureus*	*E. coli*	*E. carotovora*
DIC (mm)	IP-1	2.68 ± 0.76	1.44 ± 0.41	3.02 ± 0.32	3.94 ± 0.09
	IP-2	7.33 ± 0.25	6.96 ± 0.44	4.32 ± 0.97	5.36 ± 0.16
	IP-3	8.57 ± 0.83	8.06 ± 0.46	3.87 ± 0.82	5.85 ± 0.53
	EP-1	6.85 ± 0.19	2.19 ± 0.30	3.03 ± 0.05	4.04 ± 0.17
	EP-2	3.52 ± 0.09	1.10 ± 0.05	3.19 ± 0.39	4.42 ± 0.82
	CK	—	—	—	—
	Kantrex	24.41 ± 0.682	21.83 ± 0.528	12.14 ± 0.795	24.56 ± 1.042
MIC (mg/mL)	IP-1	5.000	10.000	2.500	2.500
	IP-2	0.156	0.156	1.250	0.625
	IP-3	0.156	0.156	2.500	0.625
	EP-1	0.313	5.000	2.500	1.250
	EP-2	2.500	10.000	2.500	1.250

Note: DIC, diameter of inhibition circle; MIC, minimum inhibitory concentration.

## Data Availability

The data that support the findings of this study are available from the corresponding author upon reasonable request.

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
