# Peer review of "Structure and Bioactivity of Intracellular and Extracellular Polysaccharides of Trametes lactinea Mycelium"

_microorganisms, 2024, doi:10.3390/microorganisms12071431_

Round 1
Reviewer 1 Report
Comments and Suggestions for Authors
Comments for authors
The paper entitled “Structure and bioactivity of intracellular and extracellular pol- 2 ysaccharides of Trametes lactinea mycelium” is very interesting research; however, it is necessary to adjust throughout the text:
1. The description of the polysaccharides with respect to the fungal background is adequate, but the way of writing is not. Line 57 starts it indicating that the results suggest, it is not correct, the text should be paraphrased since the way it is presented generates the confusion that they are results of the work submitted.
2. Page 2, line 78-80. When you indicate that biological activity depends on factors such as molecular weight, monosaccharide composition, can you indicate which ones, it would be very relevant to the work.
3. It is desirable to indicate the objective of the work at the end of the introduction.
4. Page 3, line 110-11. What was the reason for using this culture medium for the isolation, bibliographic references are needed (similar case for 2.3 section LF). Is the term natural pH correct, what does it refer to?
5. Line 138. Error in the reference.
6. Line 147. The centrifugation could be referred in x g units.
7. Evaporation at 70 °C does not damage the polysaccharides, how did you ensure that there was no modification at this temperature?
8. In the section 2.5.3. What were the chromatography conditions, what was the flow rate and how did you control it, what was the mobile phase and why, how did you collect the sample, how did you detect it as it eluted from the column, describe the method in detail so that anyone can replicate it
9. Discussion section. You need to discuss the results in depth. For example, you should indicate the importance of whether a polysaccharide has a triple helix or not. How do these structural features affect bioactivity, this is very relevant since the first words of the title include them? Same case for molecular weight, is there any correlation with bioactivity?
10. The conclusion should be restructured, since it is irrelevant to summarize the results; indicating the most important findings can make the document more attractive.
11. More than 40% of the references are outdated because this information shows old research scarce of novelty.
1
Author Response
Dear Reviewer,
Thank you very much for taking the time to review this manuscript. Please find the detailed
responses below and the corresponding revisions/corrections highlighted/in track changes in
the re-submitted files.
Comments 1: The description of the polysaccharides with respect to the fungal background is adequate, but the way of writing is not. Line 57 starts it indicating that the results suggest, it is not correct, the text should be paraphrased since the way it is presented generates the confusion that they are results of the work submitted.
Response 1: We have changed this misleading statement.
Comments 2: Page 2, line 78-80. When you indicate that biological activity depends on factors such as molecular weight, monosaccharide composition, can you indicate which ones, it would be very relevant to the work.
Response 2: We added that the biological activity affected by molecular mass refers to polysaccharide oxidative properties. And the effect of these factors on bioactivity is discussed in the Discussion section.
Comments 3: It is desirable to indicate the objective of the work at the end of the introduction.
Response 3: We have stated the objectives of our work at the end of the introduction.
Comments 4: Page 3, line 110-11. What was the reason for using this culture medium for the isolation, bibliographic references are needed (similar case for 2.3 section LF). Is the term natural pH correct, what does it refer to?
Response 4: We have added references. And we would like to express the meaning of "ph nature". We have corrected the misrepresentation.
Comments 5: Line 138. Error in the reference
Response 5: We have corrected the error in the reference.
Comments 6: Line 147. The centrifugation could be referred in x g units.
Response 6: We would like the expression about centrifugation to be contextualised so that it is easy for the reader to read.
Comments 7: Evaporation at 70 °C does not damage the polysaccharides, how did you ensure that there was no modification at this temperature?
Response 7: In our team's research on the Trametes lactinea polysaccharide extraction process, we explored and compared several evaporation temperatures and finally came up with the optimal extraction temperature of 70°C. We have referenced it in the manuscript.(Reference [16]:Liu, Y.; Shen, L.; Yang, M.; Yang, K.; Cheng, F. Extraction and chemical composition analyses of intracellular and extracellular polysaccharides from Trametes lactinealiquid fermentation. Fermentation. 2024, 10, 13. https://doi.org/10.3390/FERMENTATION10020076.)
Comments 8: In the section 2.5.3. What were the chromatography conditions, what was the flow rate and how did you control it, what was the mobile phase and why, how did you collect the sample, how did you detect it as it eluted from the column, describe the method in detail so that anyone can replicate it
Response 8: Thank you very much for your comments, about this part of the polysaccharide isolation and purification method we described in detail in this article in reference [16], and we would like to describe it in detail in this study, but another reviewer thought that the manuscript contains some duplicate data which can be found in previous reports, especially in reference [16]. he hoped that we should eliminate the duplicate data and shorten the the length of the manuscript, emphasise new data, and consider previous data. After consideration we agreed that the reviewer's comments were correct, so we have deleted the expression that duplicates reference 16 and cited it. The present study is identical to the study of reference 16, with identical samples and research methods.
Comments 9: Discussion section. You need to discuss the results in depth. For example, you should indicate the importance of whether a polysaccharide has a triple helix or not. How do these structural features affect bioactivity, this is very relevant since the first words of the title include them? Same case for molecular weight, is there any correlation with bioactivity?
Response 9: Thank you very much for your valuable suggestions, which were very useful for our article, and we have revised and adjusted the structure and presentation of the discussion section.
Comments 10: The conclusion should be restructured, since it is irrelevant to summarize the results; indicating the most important findings can make the document more attractive.
Response 10: We have modified the conclusions section to streamline the presentation and focus on the description of the relationship between polysaccharide structure and bioactivity.
Comments 11: More than 40% of the references are outdated because this information shows old research scarce of novelty.
Response 11: We have updated some of the outdated references.
Finally, Thank you again for your positive comments on our manuscript. Please let us know if you have any comments on the revision and we will be happy to consider them and revise them!
Reviewer 2 Report
Comments and Suggestions for Authors
This relatively long manuscript contains relevant information about the main polysaccharides found in liquid fermented mycelia of Trametes lactinea, focused to their applications and possible biological activities.
This is valuable, but the main concern is that the manuscript contains some duplicate data that can be found in previous reports, especially in ref. 16 (recently published by the same group and mostly related to the structure and composition of those polysaccharides) and ref. 26 (related to the ROS scavenging and antioxidant activity of those polymeric carbohydrates).
In an eventual revision, authors should eliminate duplicate data and reduce the length of the manuscript, emphasizing the novel data and discussing them in a global style, considering previous data.
For instance, the isolation of the 5 polysaccharides (3 intra- and 2 extracellular) using DEAE cellulose anion exchange and dextran Gel G-100 chromatographies has been already described. Methods about those protocols in paragraphs 2.5.2. and 2.5.3 should be reduced, and attention would be paid to reference 16.
Particular points
Abstract: Concerning the conformation (alpha, beta) of the glycosidic bonds.... what about IP-2 and IP-3? The abstract reports that IP-3 has the most potent immunomodulatory effect, so the structure of that component would be important.
Line 47: Corynebacterium macrocephala should be written in italics
Line 138, add a closing bracket to [20.
Line 213: Replace addad by added
Line 247 and 248: Replace DEME by DMEM (supposedly, the standard culture medium)
Line 298: vibration of Se-O bonds. The origin of selenium in the polysaccharides and the location of those bonds would be justified.
Figure 2 (b): Transmittance values about 100% would be justified, and the sharp peak around 1350 cm-1 would be discussed.
Line 306: 3.1.3. The dependence of the triple helical structure with basic pH (NaOH concentration) would be briefly presented or alternatively, referenced. The use of Congo Red as control or reference molecule would be also referenced.
Line 369: Table 2 contains supplementary material, but IC50, everything is accessory. That Table could be easily abbreviated.
Figure 7: The % of cell relative survival after LPS treatment would be justified.
Discussion is too long, and this section is not appropriate for introducing data or subsections. Composition of monosaccharides at Table 8 of ref. 16 and Tab 4 of this manuscript is identical. It is essential that data duplication should be avoided. The presence of this Table at the discussion is not acceptable. Conclusion is also too long. It seems an alternative abstract/summary rather than a conclusion.
Author Response
Dear Reviewer,
Thank you very much for taking the time to review this manuscript. Please find the detailed
responses below and the corresponding revisions/corrections highlighted/in track changes in
the re-submitted files.
We have removed duplicate data from the article and cited reference [16]. and have revised the discussion section and conclusion section.The following is a response to your making specific suggestions.
Comments 1: Abstract: Concerning the conformation (alpha, beta) of the glycosidic bonds.... what about IP-2 and IP-3? The abstract reports that IP-3 has the most potent immunomodulatory effect, so the structure of that component would be important.
Response 1:We have already emphasised the conformation and composition of IP-3 in the abstract section.
Comments 2: Line 47: Corynebacterium macrocephala should be written in italics. Line 138, add a closing bracket to [20. Line 213: Replace addad by added. Line 247 and 248: Replace DEME by DMEM (supposedly, the standard culture medium). Line 306: 3.1.3. The dependence of the triple helical structure with basic pH (NaOH concentration) would be briefly presented or alternatively, referenced. The use of Congo Red as control or reference molecule would be also referenced. Line 369: Table 2 contains supplementary material, but IC50, everything is accessory. That Table could be easily abbreviated.
Response 2: We have corrected these formatting and writing issues as per your comments.
Comments 3: Line 298: vibration of Se-O bonds. The origin of selenium in the polysaccharides and the location of those bonds would be justified. Figure 2 (b): Transmittance values about 100% would be justified, and the sharp peak around 1350 cm-1 would be discussed.
Response 3: After reviewing the literature again, we noted that the experimental samples in the previously referenced literature were from selenium-enriched environments, and that the samples obtained after isolation and purification in this experiment did not contain selenium. We have removed this error and discussed the sharp peak around 1350 cm-1.
Finally, thanks again for your suggestions as well as your comments. Please let us know if you have any comments on the revision and we will be happy to consider them and revise them!
Round 2
Reviewer 2 Report
Comments and Suggestions for Authors
I appreciate the style of reply letter and the predisposition of authors to consider my former suggestions useful for improving their paper. According to that, I have some new suggestions for the modified version of the manuscript.
Abstract:
Concerning the new text (lines 16-19): “The results showed that IPs have the glycopyranose ring structure. IP-3 was the only component with a triple helix structure, while the other four components did not possess this structure. IP3 has higher molecular weight, glucuronic acid content, flavonoid and total phenolic content compared to other components”,
Please consider the following considerations:
a) Assumed that IP-3 contains flavonoids and other phenols, this molecule is a conjugated polysaccharide. This type of compounds is called flavonoid-grafted polysaccharides (see doi.org/10.1016/j.ijbiomac.2018.05.149). This is important and would be mentioned instead of just “polysaccharide”. The bioactivity of the molecule, mostly antioxidant properties, would depend on the flavonoid/phenolic moiety rather than the glucuronic component.
b) Assumed that the polymer contains glucuronic, the presence of a glycopyranose ring is obvious. I suggest that the first part of the added sentence would be eliminated.
Please, explain the meaning of the expression “pH-natural” (lines 109 and 120). If possible, give the numerical value.
Lines 276-278: Assuming the presence of glucuronic acid in the polysaccharide backbone, it is surprising that the IR band at 1,400 cm-1 generated by the stretching vibration of C-O in -COOH suggest small amounts of protein or polyphenols. Glucuronate contains carboxyl groups without any other requirement. In turn, some polyphenols contain carboxyl groups, but not all. I understand that the assignation of bands to particular groups in complex conjugated biomolecules, but this assignation is particularly unnecessary and probably wrong.
Discussion and conclusion are largely re-written. I suggest that the reference mentioned above, and the expression flavonoid-grafted polysaccharides would be used in these sections. I am not author of this reference and according to the instructions, I am not trying to get self-cites of my articles, but I think that authors should read that article.
Thank you.
Author Response
Dear Reviewer,
Thank you very much for taking the time to review this re-submitted manuscript. Please find the detailed responses below and the corresponding revisions/corrections highlighted/in track changes in the re-submitted files.
Comments a): Assumed that IP-3 contains flavonoids and other phenols, this molecule is a conjugated polysaccharide. This type of compounds is called flavonoid-grafted polysaccharides (see doi.org/10.1016/j.ijbiomac.2018.05.149). This is important and would be mentioned instead of just “polysaccharide”. The bioactivity of the molecule, mostly antioxidant properties, would depend on the flavonoid/phenolic moiety rather than the glucuronic component.
Response a): Thank you for recommending the article, it was very helpful in our research. We agree with your comments and the findings of this article. It is an important reference for our conclusions. We have read the article and refer to it to modify our discussion and conclusions accordingly(Lines489-493).
Comments b):Assumed that the polymer contains glucuronic, the presence of a glycopyranose ring is obvious. I suggest that the first part of the added sentence would be eliminated. Please, explain the meaning of the expression “pH-natural” (lines 109 and 120). If possible, give the numerical value.
Lines 276-278: Assuming the presence of glucuronic acid in the polysaccharide backbone, it is surprising that the IR band at 1,400 cm-1 generated by the stretching vibration of C-O in -COOH suggest small amounts of protein or polyphenols. Glucuronate contains carboxyl groups without any other requirement. In turn, some polyphenols contain carboxyl groups, but not all. I understand that the assignation of bands to particular groups in complex conjugated biomolecules, but this assignation is particularly unnecessary and probably wrong.
Response b): We agreed with you and removed it. Regarding "pH-natural", our intention is to express that there is no need to adjust the pH value during the preparation of microbiological media. In the Chinese expression, it has the meaning of keeping pH in a natural state, but we noticed that "pH-natural" seems to cause misunderstanding to the readers, so we decided to delete the expression which is easy to cause misunderstanding. We removed this inference of unnecessary assignation(Lines 276-278)
Finally, thanks again for the recommended article and the expert suggestions! Your suggestions have helped us refine our research.